# TORSIONAL-GFN: A CONDITIONAL CONFORMATION GENERATOR FOR SMALL MOLECULES

## ABSTRACT

Generating stable molecular conformations is crucial in several drug discovery applications, such as estimating the binding affinity of a molecule to a target. Recently, generative machine learning methods have emerged as a promising, more efficient method than molecular dynamics for sampling of conformations from the Boltzmann distribution. In this paper, we introduce Torsional-GFN, a conditional GFlowNet specifically designed to sample conformations of molecules proportionally to their Boltzmann distribution, using only a reward function as training signal. Conditioned on a molecular graph and its local structure (bond lengths and angles), Torsional-GFN samples rotations of its torsion angles. Our results demonstrate that Torsional-GFN is able to sample conformations approximately proportional to the Boltzmann distribution for multiple molecules with a single model, and allows for zero-shot generalization to unseen bond lengths and angles coming from the MD simulations for such molecules. Our work presents a promising avenue for scaling the proposed approach to larger molecular systems, achieving zero-shot generalization to unseen molecules, and including the generation of the local structure into the GFlowNet model.

## 1 INTRODUCTION

Sampling molecular conformations proportionally to the Boltzmann distribution is a cornerstone of many applications in chemistry and biology. For a molecular graph $G$, this problem corresponds to generating independent molecular conformations $c \in C_G$ from the Boltzmann distribution $p(c) \propto \exp(-E(c)/k_bT)$ with known but unnormalized density. For thermodynamic calculations, a more complete sampling of the conformations provides a better estimate of important quantities in drug discovery, such as free energy, and binding affinity to a target (Molani & Cho, 2024).

Among computational chemistry methods, molecular dynamics is the *de facto* approach, where methods such as CREST have demonstrated the ability to accurately capture low-energy conformations (Pracht et al., 2020). However, despite the accuracy of these methods, it remains computationally expensive for high throughput applications and large compounds. Although there is ongoing research in faster alternatives with knowledge-based algorithms, such as distance-geometry approaches like ETKDG (Riniker & Landrum, 2015), these methods cannot sample in accordance to the Boltzmann distribution.

Recently, generative machine learning has emerged as a promising approach for conditional sampling of conformations from the Boltzmann distribution. Unlike molecular dynamics and classical sampling techniques such as Markov chain Monte Carlo algorithms, machine learning methods can potentially amortize the computational cost of sampling conformations and energy evaluations and generalize to unseen regions of the energy landscape and novel molecular systems.

For instance, Boltzmann generators (Noé et al., 2019) train normalizing flows which map from easy-to-sample distributions, like standard Gaussians, to the energy landscape of interest. Training typically consists of both data- and energy-based training. Data-based training involves maximizing the likelihood of the observed data (most often using the forward-KL loss), while energy-based training minimizes the reverse-KL loss between the flow's learned density and the unnormalized target Boltzmann density; however, reverse-KL and forward-KL formulations typically suffer from mode-seeking and mean-seeking behaviors, respectively, which can adversely impacting training dynamics and sample quality (Malkin et al., 2022). Although learning with off-policy distributions is

possible with re-weighted importance sampling, it often introduces a high gradient variance. Recently, GFlowNets (Bengio et al., 2023) have emerged as a new class of generative models that can be trained to sample according to a reward function. Further, they can be trained using any behavior policy without introducing high gradient variance (Malkin et al., 2022), making them well-suited for the problem of exploring the energy landscape of molecules.

In this paper, we introduce Torsional-GFN, a method to sample conformations of a molecule conditioned on the molecular graph and the local structure (i.e. bond lengths and angles). Torsional-GFN is a conditional GFlowNet that generates conformations of a given molecular graph by sampling rotations of its torsion angles. We summarize the contributions of our work as follows:

- We demonstrate the ability of a single GFlowNet model to approximate the Boltzmann distribution of multiple molecules, extending previous approaches (Volokhova et al., 2024) which required training one GFlowNet per molecular system,
- we introduce a new graph neural network architecture, VectorGNN, designed to infer geometrical information about the torsion angles in a molecule's 3D structure,
- we demonstrate the ability of GFlowNets, trained on a dataset of six molecules with fixed bond lengths and angles, to generalize to unseen values of the bond lengths and angles from MD simulations of new molecules.

## 2 RELATED WORK

This section reviews the most relevant literature to this work. In particular, we provide additional context about how molecular dynamics (MD) is used to sample molecular conformations and we review alternatives to it, including Boltzmann generators, diffusion models, and in particular methods that leverage the representation of molecular configurations in terms of torsion angles.

**Molecular dynamics (MD) simulations**  MD consists of running simulations of the evolution of the molecule's position over time using Newtonian physics. If we let MD run for a sufficient amount of time, then a Boltzmann-weighted ensemble of conformations can be obtained directly. This allows for the calculation of any desired property of a molecule; however, to obtain uncorrelated samples from MD while maintaining stability, small update steps, on the order of femtoseconds, are typically needed. This makes MD simulations very slow, and more challenging when we have well-separated metastable states, where transitions are unlikely due to high-energy barriers (Klein & Noé, 2024).

**Boltzmann generators**  In (Noé et al., 2019), the authors tackle the problem of sampling from an unnormalized Boltzmann density using normalizing flows. Training consists of both data-based and energy-based training, using respectively forward-KL loss and reverse-KL loss. However, these losses suffer respectively from mode-seeking and mean-seeking behaviors, thus hurting training (Malkin et al., 2022). Learning with off-policy distributions is possible with reweighted importance sampling, but it introduces a high gradient variance. The GFlowNet trajectory-based losses encompass these issues, and they can be trained using any behavior policy without introducing high gradient variance (Malkin et al., 2022).

**Diffusion models trained with data**  After the success of diffusion models for images, several works have extended it for molecular conformations sampling, using a predefined dataset of low-energy molecular conformations and / or MD trajectories. For instance, (Jing et al., 2022), a conditional diffusion model is trained to sample conformations of a molecule given its molecular graph as an input. To make the conformer search problem more manageable, authors apply a widely-used approximation for small molecules, called the rigid rotor (RR) approximation (Hawkins, 2017). In this framework, the bond lengths and bond angles are kept fixed, and the only degree of freedom is the torsion angles. TorsionalDiffusion is trained on QM9 and GEOM-DRUGS (Axelrod & Gomez-Bombarelli, 2022), which provides standard conformer ensembles generated with metadynamics in CREST (Pracht et al., 2020). Besides training with data, the authors propose an energy-based training approach to learn to sample from the corresponding Boltzmann distribution. However, this approach relies on reweighted importance sampling, yielding a variance that increases with the discrepancy between the learned policy and the true Boltzmann distribution (Malkin et al., 2022).

**Diffusion models trained with energy**    Diffusion processes can be trained to sample from a target distribution without necessarily having access to data. For example, Akhound-Sadegh et al. (2024) introduce an offline, simulation-free score matching loss, relying solely on the energy and its gradients to get an estimation of the true score function. This loss can be computed offline; yet it is biased, and it requires a large number of energy evaluations per gradient update to get a reasonable estimation. The authors show results on systems up to the 55-particle Lennard-Jones system. More recently, Havens et al. (2025) train a conditional diffusion sampler on both SPICE and GEOM-DRUGS datasets. Starting from a stochastic optimal control framework, they derive an on-policy regression loss that matches the learned vector field to the gradient of energies. Using a replay buffer allows them to significantly decrease the number of required energy calls per gradient update compared to Havens et al. (2025).

In this paper, we stray away from the diffusion framework, for which the noising process is fixed prior to training. Unlike diffusion models, GflowNets allow for learning the noising process. In addition, their loss allows for combining on-policy and off-policy learning, thus providing a nice balance between exploitation and exploration.

**GFlowNets for molecular conformation generation**    Lahlou et al. (2023); Volokhova et al. (2024) introduced GFlowNets for sampling conformations of small molecules from the Boltzmann distribution. In these earlier works, a dedicated GFlowNet with a MLP policy model was trained for each molecule to learn the potential energy surface. Here, we build upon these studies by introducing a conditional GFlowNet with a novel GNN architecture that serves as a policy model taking in the molecular graph as input. Moreover, we establish a framework for training a single GFlowNet on multiple molecules, potentially enabling amortized inference.

## 3    METHOD

Here, we introduce the main contributions of this work, namely the Torsional-GFN method and the VectorGNN architecture that makes the policy model of Torsional-GFN. The general pipeline is illustrated in figure 3.

### 3.1    PRELIMINARIES

Adopting the definitions and notations from Jing et al. (2022), we consider a molecular graph $G = (V, E)$, where nodes $V$ are the atoms and edges $E$ are the atom bonds. $C_G$ denotes the space of possible conformations.

A conformation $c \in C_G$ is a set of SE(3)-equivalent vectors in $\mathbb{R}^{3|V|}$. In other words, $C_G$ maps to the quotient space $\mathbb{R}^{3|V|}/\text{SE}(3)$. In this space, two vectors that are equal up to an SE(3)-transformation belong to the same equivalence class, and are therefore considered to be the same conformation.

Our goal is to draw independent conformations $c \in C_G$ from the Boltzmann distribution:

$$p(c|G) = \frac{1}{Z(G)} \exp\left(\frac{-E(c)}{k_b T}\right), \tag{1}$$

where $E(c)$ is the internal energy of conformation $c$, $Z(G)$ is a normalization constant for the molecular graph $G$, $k_B$ is the Boltzmann constant, and $T$ is the absolute temperature.

The definitions above describe the space of conformations in terms of an extrinsic (or Cartesian) coordinate frame. A conformation $c \in C_G$ can also be specified in terms of its *intrinsic coordinates*:

- Bond lengths and bond angles, referred to as the molecule's local structure, are denoted by $L$,
- Torsion angles, referred to the dihedral angles around freely rotatable bonds, are denoted by $\Phi$ (see Appendix A.2.1).

Previous works (Volokhova et al., 2024; Jing et al., 2022) either rely on the assumption that the local structure of stable conformations is constant (Hawkins, 2017; Riniker & Landrum, 2015), or they sample local structures from RDKit cheminformatics software . However, we here observe that, if we care about the full potential energy landscape, such approximations do not hold, as shown in

Appendix A.3.4 on a dataset of 8 molecules. In particular, RDKit significantly underestimates the bond lengths vibrations (Figure 5a).

## 3.2 TORSIONAL-GFN

Given a molecular graph $G$, we construct 3D conformations $c \in C_G$ by sampling both the local structure $L \sim p(L|G)$ and the rotatable torsion angles $\Phi \sim p(\Phi|L, G)$. This allows us to write $c = c(L, \Phi)$, making the intrinsic coordinates explicit, and to decompose the probability density from Equation (1) into

$$p(c|G) = \underbrace{\frac{1}{\sqrt{\det(g)}} p(\Phi|G, L)}_{\text{learned by Torsional-GFN}} \underbrace{p(L|G)}_{\text{sampled from MD}}, \tag{2}$$

where $\frac{1}{\sqrt{\det(g)}}$ describes the volume change due to the conversion from intrinsic to extrinsic coordinates. We set $\det(g) = 1$ for simplicity and offer a detailed discussion in Appendix A.2.2.

To reduce the dimensionality of our search space, following Volokhova et al. (2024), we let the GFlowNet learn the distribution of torsional angles $p(\Phi|G, L)$ and obtain the local structures $p(L|G)$ from MD simulations. Note that $p(\Phi|G, L) = \frac{1}{Z(G,L)} \exp\left(-\frac{E(c(\Phi,L))}{k_B T}\right)$, where $Z(G, L)$ is a normalization constant, which does not depend on $\Phi$. As such, we can define the *unnormalized* reward function for the vector of torsion angles $\Phi$, conditioned on $G$ and $L$, as:

$$R(\Phi|G, L) = \exp\left(-\frac{E(c(\Phi, L))}{k_B T}\right). \tag{3}$$

We train a GFlowNet model, which is designed to learn sampling probabilities $p_\top^\theta(\Phi|G, L)$ proportional to the reward in Equation (3).

Torsional-GFN is a continuous GFlowNet (Lahlou et al., 2023), conditioned on molecules defined by their molecular graph and local structure $(G, L)$. Given a molecule with $m$ rotatable torsion angles, $\Phi = (\phi^1, \dots, \phi^m)$, Torsional-GFN samples these torsion angles in the hypertorus $\mathcal{X} = [0, 2\pi]^m$ (Volokhova et al., 2024). The sampling process starts from a source state $\Phi_0$ and continues with a trajectory of sequential updates $\tau = (\Phi_0 \to \Phi_1 \to \cdots \to \Phi_n = \Phi)$ according to a trainable forward policy $P_F(\Phi_t|\Phi_{t-1})$. Torsional-GFN also uses a trainable backward policy $P_B(\Phi_{t-1}|\Phi_t)$. Similarly as in Volokhova et al. (2024), we parametrize the forward and backward policies of Torsional-GFN as a mixture of von Mises distributions with learnable mixing weights $w$, locations $\mu$ and concentrations $\kappa$ (details in Appendix A.2.3).

Given a batch of trajectories $B_\tau$ that can be sampled using any behavior policy, we train Torsional-GFN to minimize the Vargrad loss (Richter et al., 2020):

$$\mathcal{L}_{\text{VG}}(B_\tau; \theta|G, L) = \mathbb{E}_{\tau \in B_\tau} \left(\log \mathcal{C}_\theta(\tau|G, L) - \mathbb{E}_{\tau' \in B_\tau} \log \mathcal{C}_\theta(\tau'|G, L)\right)^2$$

$$\text{where} \quad \mathcal{C}_\theta(\tau|G, L) = \frac{P_B^\theta(\tau \mid \Phi_n, G, L) R(\Phi_n|G, L)}{P_F^\theta(\tau|G, L)}$$

Thus, we can train Torsional-GFN on a dataset $D$ of molecular graphs with their local structures using the following loss:

$$\mathcal{L}_D = \mathbb{E}_{G_i, L_i \sim D} \mathcal{L}_{\text{VG}}(B_\tau; \theta|G_i, L_i). \tag{4}$$

Note that the behavior policy, used to sample trajectories and take gradient steps on $\mathcal{L}_D$ can be *any full-support off-policy distribution* (see Algorithm 1), unlike in Boltzmann generators (Noé et al., 2019), where the behavior policy is restricted to either the forward or reverse-KL loss. Moreover, Torsional-GFN does not require any importance sampling (Jing et al., 2022), thus avoiding the issues of bias and high-variance that come with it.

### 3.3 VECTORGNN

In order to train a single conditional GFlowNet with multiple molecular systems—unlike previous work (Volokhova et al., 2024), where a separate GFlowNet was trained for each molecular system with a policy parametrized with an MLP—we introduce a new graph neural network architecture, VectorGNN, for the Torsional-GFN policy models ($P_F^\theta$ and $P_B^\theta$). We found VectorGNN to be faster than other GNNs such as the one used in Jing et al. (2022).

A mixture of von Mises distributions with components $k \in 1, \ldots K$ is defined with locations $\mu_k(\phi)$, concentrations $\kappa_k(\phi)$ and mixture weights $w_k(\phi)$. The location parameter determines the expected direction of torsion angle updates, and must thus be *reflection-equivariant*. VectorGNN learns forces that generate a torque for each rotatable bond, then outputs a set of pseudo-scalars $o_\phi \in \mathbb{R}^{K \times 3}$, which are are mapped to the distribution parameters as follows:

$$w_k(\phi) = (o_\phi)^2_{k,1}, \quad \mu_k(\phi) = (o_\phi)_{k,2}, \quad \kappa_k(\phi) = (o_\phi)^2_{k,3}.$$

**Invariant Message Passing**  The first $L$ layers of VectorGNN perform invariant message passing over a fully-connected molecular graph to compute atomic embeddings:

$$m_i^l = \sum_{j=1}^{N} \mathrm{MLP}_m(h_i^l, h_j^l, e_{ij}, \|\vec{x}_{ij}\|), \tag{5}$$

$$h_i^{l+1} = \mathrm{MLP}_h(h_i^l, m_i^l), \quad h_i^{l+1} \in \mathbb{R}^D, \tag{6}$$

where $\vec{x}_{ij}$ is the vector between atoms $i$ and $j$, and $e_{ij}$ denotes edge attributes. The initial atomic embeddings $h_i^0$ consist of one-hot encodings of the atom type, atomic number, one-hot encoding of the atom degree, and one-hot encoding of the atom hybridization. The edge attribute $e_{ij}$ is one-hot encoding of the bond type. Here, MLP denotes a multi-layer perceptron with GELU activations (Hendrycks & Gimpel, 2016).

**Reflection-equivariant Output**  The learned invariant embeddings are used to predict the amplitudes of interatomic forces between all atom pairs:

$$f_{ij} = \mathrm{MLP}_f(h_i^L + h_j^L, e_{ij}, \|\vec{x}_{ij}\|) \in \mathbb{R}^{C \times 3}, \tag{7}$$

where $C$ is the number of force channels. These amplitudes are used to compute directional interatomic force vectors:

$$\vec{f}_{ij} = \frac{f_{ij}\vec{x}_{ij}}{\|\vec{x}_{ij}\|^3}, \quad \vec{f}_i = \sum_{j=1}^{N} \vec{f}_{ij},$$

where $\vec{f}_i$ is the net force acting on atom $i$.

For a given rotatable bond between atoms $i$ and $j$, the neighboring atoms $k \in \mathcal{N}(i)$ exert forces that generate torques about the bond:

$$\vec{t}_{ij} = \sum_{k \in \mathcal{N}(i),\, k \neq j} \vec{x}_{ik} \times \vec{f}_k. \tag{8}$$

Analogously, we compute $\vec{t}_{ji}$, the torque induced at atom $j$, which acts in the opposing direction to $\vec{t}_{ij}$. Since we are only interested in rotations around the bond axis, we project both torque vectors onto the bond direction and compute the net torque magnitude $s_{ij} \in \mathbb{R}^C$ as:

$$t_{ij} = \vec{t}_{ij} \cdot \frac{\vec{x}_{ij}}{\|\vec{x}_{ij}\|}, \quad s_{ij} = t_{ij} - t_{ji}. \tag{9}$$

The resulting pseudo-scalar torque magnitudes $s_{ij}$ are then passed through a reflection-equivariant MLP to produce the final output features:

$$o_{ij} = \mathrm{EquivMLP}(s_{ij}) \in \mathbb{R}^{K \times 3}, \tag{10}$$

$$\mathrm{EquivMLP}(x) = W_1 \sin(W_2 x). \tag{11}$$

Because $o_{ij} = -o_{ji}$, each rotatable bond $\phi$ can be represented using either directed edge $ij$ or $ji$.

The first $K$ elements of $o_{ij}$ are passed to a softmax layer to predict the weights, the next $K$ elements correspond to the predicted locations, and the absolute value of the last $K$ elements predict the concentrations.

# 4 EXPERIMENTS AND RESULTS

In order to evaluate the performance of Torsional-GFN, we train the model using a set of six molecules as conditions, each with $m = 2$ rotatable torsion angles, from the FreeSolv dataset (Mobley & Guthrie, 2014). To investigate generalization to unseen molecules, we use two additional molecules as a test set . The first transition $\Phi_0 \rightarrow \Phi_1$ is sampled uniformly on $[0, 2\pi]^m$, and is followed by $n = 5$ transitions from the learned mixture of von Mises distributions.

For evaluation, we run MD simulations to obtain a dataset of 2001 ground truth conformations from the Boltzmann distribution for each molecule. During training, bond lengths and angles for each molecule were fixed to the values of only one arbitrary conformation from the MD simulation dataset. We provide more detail in Appendix A.3.2.

As energy function for the reward of Torsional-GFN, we use MMFF94s (Halgren, 1999), a widely-used force field that provides a good estimation of the internal energy for small organic molecules.

In order to accelerate the training process, we pre-train VectorGNN on the supervised task of predicting the energy and torsion angles values of molecular conformations, as detailed in Appendix A.3.3. Then, we use the pretrained model as an initialization for training both $P_F^\theta$ and $P_B^\theta$.

Our main objective for the evaluation is to investigate how closely the torsion angles from Torsional-GFN align with those from the true Boltzmann distribution, and whether the model can generalize to unseen local structures and unseen molecules.

## 4.1 PROXIMITY TO THE REWARD LANDSCAPE

At convergence, the sampling distribution of the Torsional-GFN $p_\top^\theta(\Phi|G, L)$ should be proportional to the reward distribution $R(\Phi|G, L)$. To test this, we look at the correlation between these two quantities (in the log scale) on points sampled from the torus: we discretize the 2D torus to a grid of 10,000 points $\{\Phi_i\}_{i=1}^{10000} \in [0, 2\pi]^2$. Then, we estimate $\log p_\top^\theta(\Phi_i|G, L)$ via importance sampling as done by Volokhova et al. (2024), and compute its correlation with $\log R(\Phi_i|G, L)$ on the grid. We refer to this metric as $\rho_{\log p_\top^\theta, \log R}$ in Table 1. Additional details are provided in Appendix A.3.5.

In addition, we estimate the discretized target distribution $P(\Phi_i|G, L) = R(\Phi_i|L, G)/\sum_k R(\Phi_k|L, G)$, and the discretized sampling distribution $P_\top^\theta(\Phi_i|G, L) = p_\top^\theta(\Phi_i|L, G)/\sum_k p_\top^\theta(\Phi_k|L, G)$, and compare them using the Jenson-Shannon divergence $\mathrm{JSD}\left(P_\top^\theta(\Phi_i|G, L)||P(\Phi_i|G, L)\right)$, that we denote as $\mathrm{JSD}^P$ in Table 1.

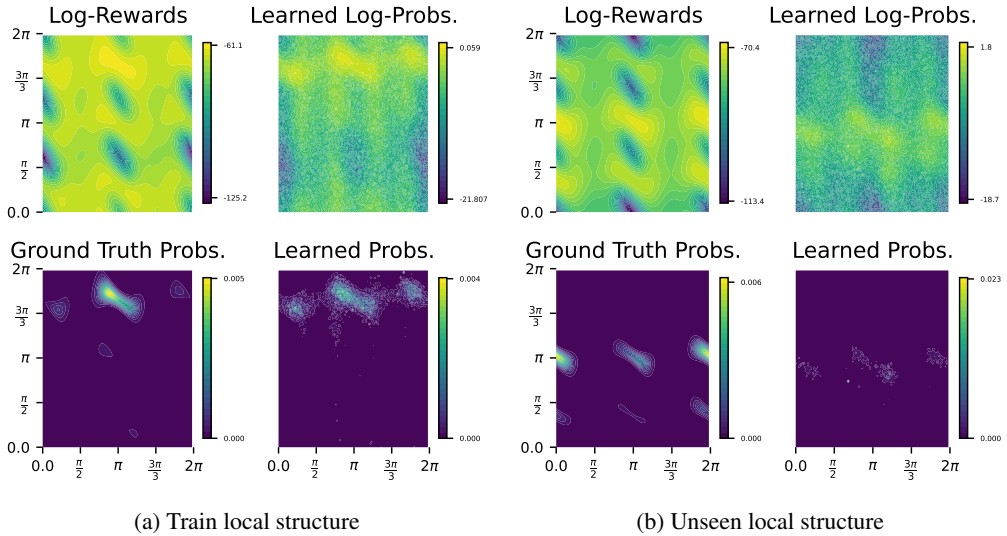

(a) Train local structure        (b) Unseen local structure

Figure 1: Visualization of the log-rewards and ground truth probabilities alongside the learned sampling distributions for molecule `COc1ccccc1`.

Figure 1 shows the log-reward landscape and the estimated log-probabilities for one of the train molecules, as well as the ground truth probabilities and estimated probabilities. Figure 1a shows the results for the local structure used during training and Figure 1b shows results for an unseen local structure. For this molecule, the metrics are close to the median value over the training set of molecules (see Figures 7 to 11 for visualizations of the results for other molecules). Table 1 provides the values of $\text{JSD}^p$ and $\rho_{\log p_T^\theta, \log R}$ for the train and unseen local structures of the 6 train and 2 test molecules.

| SMILES | $\text{JSD}^E_{\text{GFN}}/\text{JSD}^E_{rand} \downarrow$ | $\text{JSD}^p \downarrow$ | | $\rho_{\log p_T, \log R} \uparrow$ | |
| --- | --- | --- | --- | --- | --- |
| | | **Train** | **Unseen** | **Train** | **Unseen** |
| C1C=CC[C@@H]2[C@@H]1... | 0.3613 | 0.4097 | 0.6845 | 0.5493 | 0.3011 |
| COC=O | 0.0201 | 0.0358 | 0.0911 | 0.7068 | 0.7042 |
| C[C@@H]1CCCC[C@@H]1C | 0.1391 | 0.0270 | 0.2354 | 0.9561 | 0.6907 |
| COc1ccccc1 | 0.4255 | 0.1001 | 0.3484 | 0.6929 | 0.6338 |
| c1ccc2c(c1)C(=O)... | 0.0580 | 0.1417 | 0.2710 | 0.9269 | 0.9292 |
| CCC | 0.0433 | 0.0100 | 0.0256 | 0.9740 | 0.9242 |
| CCc1cccc2c1cccc2 | 0.9807 | – | 0.6206 | – | -0.0597 |
| c1ccc(c(c1)C(F)... | 0.3653 | – | 0.6795 | – | 0.3670 |

Table 1: Metrics to evaluate the proximity between the sampling and the target distributions. The top 6 rows correspond to train molecules and the bottom 2 rows are test molecules.

Both visualizations and quantitative results suggest that Torsional-GFN is able to learn the overall reward landscape for most of the train molecules and train local structures. We also observe that our model captures the main features of the landscape for the unseen local structures. By comparing Figure 1b and Figure 1a, we clearly see that Torsional-GFN is able to follow the shift of the modes in the energy landscape when sampling torsion angles for unseen local structures. Finally, in the results for the test molecules with corresponding unseen local structures (see Figure 12 and the last 2 rows of Table 1), while we observe that for one molecule our model sampling distribution is very far from the target, for the other test molecule we notice limited coverage of some modes and low probability regions of the energy landscape. This does not demonstrate consistent generalization to the test molecules, but indicates a potential for it.

## 4.2 Energy distributions

To further compare the sampling distribution with the target, we analyze the energy distribution of sampled conformations for each molecule. The target distribution here is given by the energies of the samples obtained with MD simulation. To obtain the corresponding distribution for Torsional-GFN, we sample one set of torsion angles for each local structure from MD simulation and then estimate the energy of the resulting conformations. For comparison, we also draw a random set of torsion angles from the uniform distribution on a torus. This results in three histograms per molecule, for which we restrict the energy range to the lower and upper bound of energies of MD.

To numerically measure the differences in energy distributions for each molecule, we normalize the energy histograms described above, resulting in three discrete probability distributions of the energies: $P_{MD}(E)$ for MD samples, $P_{GFN}(E)$ for Torsional-GFN samples, and $P_{rand}(E)$ for random samples. Then we compute Jenson-Shannon divergence between MD energies and Torsional-GFN energies $\text{JSD}^E_{\text{GFN}} = \text{JSD}\Big(P_{\text{MD}}(E)||P_{\text{GFN}}(E)\Big)$ and between MD energies and random energies $\text{JSD}^E_{rand} = \text{JSD}\Big(P_{\text{MD}}(E)||P_{rand}(E)\Big)$. Finally, we divide the first by the latter and obtain the ratio $\text{JSD}^E_{\text{GFN}}/\text{JSD}^E_{rand}$. If this ratio is less than 1, this implies that Torsion-GFN distribution is closer to the target than random. The first column of table Table 1 shows the values of the ratio for each of the 6 train molecules and 2 test molecules.

The values of the ratio metrics $\text{JSD}^E_{\text{GFN}}/\text{JSD}^E_{rand}$ and the energy histograms in Figure 2 show that the energy distribution of Torsional-GFN samples is significantly closer to the MD energies than random for all the train molecules and one of the test molecules. Noticeably, for several train molecules, we observe that Torsional-GFN energies histogram overlap very closely with MD energies. This may suggest that the Torsional-GFN sampling distribution is close to the target distribution for many of the unseen local structures of the train molecules, which is consistent with the evaluation comparing

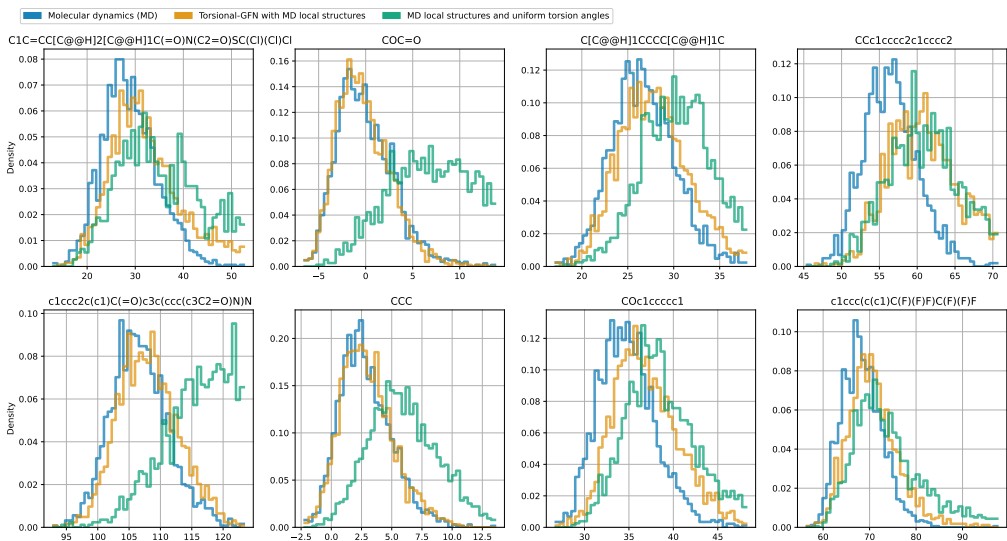

Figure 2: Energy histograms for the 6 train molecules (first three columns) and 2 test molecules (last column).

probability landscapes described earlier in this section. Similarly, we observe that Torsional-GFN energies are very close to random for one of the test molecule, while for the other test molecule the model samples noticeably closer to the MD energies than random, pointing to potential generalization capacity to unseen molecules.

## 5 CONCLUSION AND FUTURE WORK

In this work we proposed Torsional-GFN, a conditional GFlowNet method for sampling torsion angles of molecular conformations proportionally to the target distribution, and with conditioning on the input molecular graph and the local structure. Our experimental results suggest that Torsional-GFN is able to sample closely to the target distributions for the train molecules. However, some of them are more challenging to learn than others. Moreover, our method demonstrates generalization to unseen local structures and a potential for generalization to unseen molecules.

We consider this work to be an intermediate milestone towards a GFlowNet-based sampler from the Boltzmann distribution which would be able to generalize to unseen molecules. In future work, we will explore possible ways of training Torsional-GFN on a bigger dataset of molecules with more torsion angles, sampling of the local structures with the GFlownet, and different ways to parametrize the policy distribution, e.g. with a simulation-free objective (Akhound-Sadegh et al., 2024).

## IMPACT STATEMENT

The proposed method has a potential for speeding-up the sampling of small molecular conformations from the Boltzmann distribution, especially once current limitations are addressed by future work. This could affect various applications related to chemical properties prediction, such as drug discovery and materials design. We acknowledge that our work can also have potential malicious applications, such as chemical weapons development, and we firmly stand against these. Moreover, we acknowledge that if our work was used for potentially socially beneficial applications, such as drug discovery, it would be likely that only privileged big pharmaceutical companies would be able to use our method at scale and would capitalize on it, because training the Torsional-GFN is computationally expensive and requires a GPU with at least 40 GB of memory even for a small setting with 6 train molecules. Our deepest hope is that our work can contribute towards mitigating current and future pandemics, especially in underprivileged countries, which suffer the most from some classes of pandemics such

as anti-microbial resistance. However, we acknowledge that such a contribution is very challenging in the context of unequal distribution of wealth in the world.

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

# A APPENDIX

## A.1 OVERVIEW OF THE TORSIONAL-GFN PIPELINE

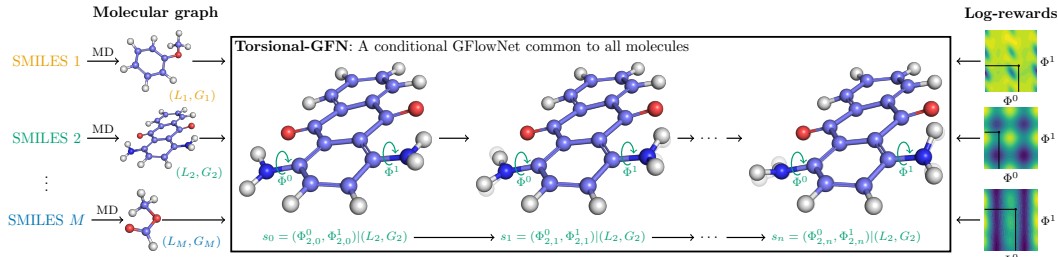

Figure 3: Overview of the Torsional-GFN pipeline

## A.2 METHOD DETAILS

### A.2.1 TORSION ANGLES PARAMETRIZATION

As there are multiple ways to define torsion angles for a molecule (Jing et al., 2022), for each rotatable bond $(b_i, c_i)$, we choose arbitarily an atom $a_i$ in the neighborhood of $b_i$, and an atom $d_i$ in the neighborhood of $c_i$. Then, we define a torsion angle of bond $b_i, c_i$ using the quadruplet $a_i, b_i, c_i, d_i$. The arbitrary choice of $a_i, d_i$ does not affect our method: another choice $(a'_i, d'_i) \neq (a_i, d_i)$ would simply result in a translation (modulo $2\pi$) of the reward landscape on the torus, making our method translation-equivariant to the choice of reference torsion angle.

### A.2.2 Likelihood conversion from torsional space to $\mathbb{R}^{3|V|}/SE(3)$

Consider a molecule described by a vector $x \in \mathbb{R}^{3|V|}/SE(3)$, with $|V| \in \mathbb{N}^+$ the number of atoms. The Boltzmann distribution is usually computed in this space of extrinsic coordinates. However, we here operate on the space of *intrinsic coordinates* $(L, \Phi)$. This change of variables implies the need to account for a factor change in the infinitesimal volume $dV$, when going from intrinsic to extrinsic space.

As we are sampling the local structure $L$ directly from MD data, we can omit the volume change due to $L$, and restrict it to the space of rotatable torsion angles $\mathcal{X} = [0, 2\pi]^m$. Note that if we were sampling $L$ with TorsionalGFN, we would need to account for the volume change due to $L$ as well.

The metric tensor $g$ is an object that describes the geometry of a coordinate system, that is its distances, angles and volumes (Physics, 2025). Its components are dot products between basis vectors. For the molecule described above, the metric tensor $g(\mathbf{x})$ is defined as:

$$g_{ij}(\mathbf{x}) = \langle \frac{d\mathbf{x}}{d\Phi_i} \cdot \frac{d\mathbf{x}}{d\Phi_j} \rangle, \quad \forall x \in \mathbb{R}^{3|V|}/SE(3), (i, j) \in [0, m-1]^2, \tag{12}$$

where

$$\frac{d\mathbf{x}}{d\Phi_i} = \begin{cases} \frac{\mathbf{x}_{b_i} - \mathbf{x}_{c_i}}{\|b_i - c_i\|} \times (\mathbf{x} - \mathbf{x}_{c_i}) & \text{if } \mathbf{x}_l \in \mathcal{V}_{c_i}, \\ 0 & \text{if } \mathbf{x}_l \in \mathcal{V}_{b_i}, \end{cases} \tag{13}$$

where $(bi, ci)$ is the freely rotatable bond for torsion angle $i$, and $\mathcal{V}_{b_i}$ is the set of all nodes on the same side of the bond as $b_i$ (Jing et al., 2022).

This allows us to compute $dV$ as follows:

$$dV = \sqrt{det(g)}d\mathbf{x}. \tag{14}$$

Thus,

$$p(\boldsymbol{x}|L, G)dx = \frac{p(\Phi|L, G)}{dV}dx = \frac{p(\Phi|L, G)}{\sqrt{det(g)}} \tag{15}$$

While it is crucial to take this factor, $\sqrt{det(g)}$, into account to sample from the Boltzmann distribution, in our experiments we set $det(g) = 1$ for simplicity. It is also omitted in the definition of the reward function for compatibility with our experiments. Future work will consider integrating this factor into the reward.

### A.2.3 Forward and backward policy parametrization

As a continuation of previous work by Volokhova et al. (2024), we parametrize the forward and backward policies of the GFlowNet as a mixture of von Mises distributions of $[0, 2\pi]^m$, that is:

$$P_F^\theta(\phi_{t+1}|\phi_t) = \sum_{k=1}^{K} w_{k,F}^\theta \cdot \text{VM}(\phi_{t+1}|\mu_{k,F}^\theta(\phi_t), \kappa_{k,F}^\theta(\phi_t)),$$

$$P_B^\theta(\phi_t|\phi_{t+1}) = \sum_{k=1}^{K} w_{k,B}^\theta \cdot \text{VM}(\phi_t|\mu_{k,B}^\theta(\phi_{t+1}), \kappa_{k,B}^\theta(\phi_{t+1})),$$

where

- VM is the von Mises distribution on the torus,
- $K$ is the number of components of the mixture,
- $w_{k,F}^\theta$ (resp. $w_{k,B}^\theta$) are the weights of the mixture (i.e. they are positive and sum to 1),
- $\mu_{k,F}^\theta(\phi)$ (resp. $\mu_{k,B}^\theta(\phi)$) is the location of the $k$-th component,
- $\kappa_{k,F}^\theta(\phi)$ (resp. $\kappa_{k,B}^\theta(\phi)$) is the concentration of the $k$-th component.

---

**Algorithm 1** TorsionalGFN training

---

**Input:** Dataset of molecular graphs and their local structures from MD: $D = \{(G_i, L_i)\}_{1 \leq i \leq |D|}$
GFlowNet initialized with arbitrary policies $P_F^\theta$ and $P_B^\theta$
Reward-prioritized ReplayBuffer (Vemgal et al., 2023) $\mathcal{R} = \{\mathcal{R}_i\}_{0 \leq i \leq |D|}$
Number of training steps $N_{steps}$

**for** $t = 1$ **to** $N_{steps}$ **do**
    Sample a batch of graphs and local structures $B = \{(G_{k_1}, L_{k_1}), \ldots, (G_{k_b}, L_{k_b})\}, \quad 1 \leq b \leq |D|$
    $\mathcal{L}_D = 0$
    **for** $j = 1$ **to** $b$ **do**
        Sample a batch of trajectories of torsion angles $B_\tau | G_{k_j}, L_{k_j}$ using the behavior policy. 50% of the trajectories are sampled using an $\epsilon$-greedy policy (Jain et al., 2022), with $\epsilon = 0.3$, and 50% are sampled by choosing a terminal state in $\mathcal{R}_{k_j}$ and constructing a trajectory backwards with $P_B^\theta$).

        Compute the VarGrad loss $\mathcal{L}_{VG}(B_\tau, \theta | G_{k_j}, L_{k_j})$

        $\mathcal{L}_D = \mathcal{L}_D + \frac{1}{b} L_{VG}(B_\tau, \theta)$

        Update $\mathcal{R}_{k_j}$ with $B_\tau$, using reward-prioritization with diversity criterion (Vemgal et al., 2023)
    **end for**

    Backprop on $\mathcal{L}_D$ and update the parameters of $P_F^\theta$ and $P_B^\theta$
**end for**

---

We fix the number of components, and learn the weights, locations and concentrations of each rotatable torsion angle in a molecule using VectorGNN, a new graph-neural network architecture which we describe in the following section. Using such a GNN allows to train a single GFlowNet model on multiple molecular systems, extending on previous work (Volokhova et al., 2024), where a separate GFlowNet was trained for each molecular system, and where the policy was parametrized with an MLP.

### A.2.4 TORSIONALGFN ALGORITHM

Algorithm 1 summarizes the TorsionalGFN training.

### A.3 EXPERIMENTS

### A.3.1 DATASET

Molecular graphs are visualized in Fig.4

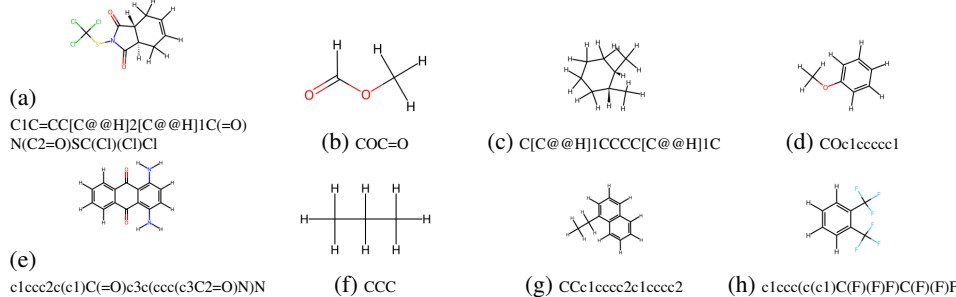

(a)
C1C=CC[C@@H]2[C@@H]1C(=O)N(C2=O)SC(Cl)(Cl)Cl

(b) COC=O

(c) C[C@@H]1CCCC[C@@H]1C

(d) COc1ccccc1

(e)
c1ccc2c(c1)C(=O)c3c(ccc(c3C2=O)N)N

(f) CCC

(g) CCc1cccc2c1cccc2

(h) c1ccc(c(c1)C(F)(F)F)C(F)(F)F

Figure 4: Visualisation of the 8 molecular graphs in the dataset

### A.3.2   MOLECULAR DYNAMICS SIMULATION

To generate conformations with Molecular Dynamics simulation, we used OpenMM (Eastman et al., 2017) with OpenFF 2.1.1 forcefields (Boothroyd et al., 2023) to compute energies in vacuum at room temperature. To get reference MD simulations we run 2ns simulations at a resolution of 1fs and subsample to get decorrelated 1ps frames. In the end, we obtain 2001 conformations for each molecule.

### A.3.3   POLICY PRE-TRAINING

We found that training a randomly initialized VectorGNN with the GFlowNet loss required considerably more iterations than the MLP policy trained by Volokhova et al. (2024). To mitigate the additional training complexity of using a GNN policy, we pre-train VectorGNN on the supervised task of predicting the energy and torsion angles values of molecular conformations. To construct a dataset for this task, we use six train molecules with one corresponding local structure from MD dataset and sample 10000 values of rotatable torsion angles from the uniform distribution on a torus $[0, 2\pi]^2$ for each molecule. This gives us 60000 conformations, which we divide into 48000 for pretraining and 12000 for evaluation of the pretrained model. The model is trained to predict energy of the conformation, $\sin \phi$, and $\cos \phi$ for all rotatable torsion angles $\phi$ of a molecule.

Then, we use the pretrained model as an initialization for training both $P_F^\theta$ and $P_B^\theta$. The the last equivariant MLP block of VectorGNN is initialized separately without pretraining. This provides an informative initialization for the backbone VectorGNN model, which has been incentivized to extract useful representations for predicting energy and shape of the conformations – a desirable inductive bias to facilitate Torsional-GFN training.

### A.3.4   COMPARISON OF THE LOCAL STRUCTURES DISTRIBUTION WITH MD SIMULATIONS AND RDKIT

The Figure 5 visualizes distribution of the values of the bond lengths and angles in the conformations sampled with MD and with RDKit. Noticeably, the distributions for MD conformations have smooth peaks with significant variance. This confirms that local structures are not fixed to specific values in the ground truth Boltzmann distribution, but follow a smooth distribution due to temperature vibrations of atoms in a molecule.

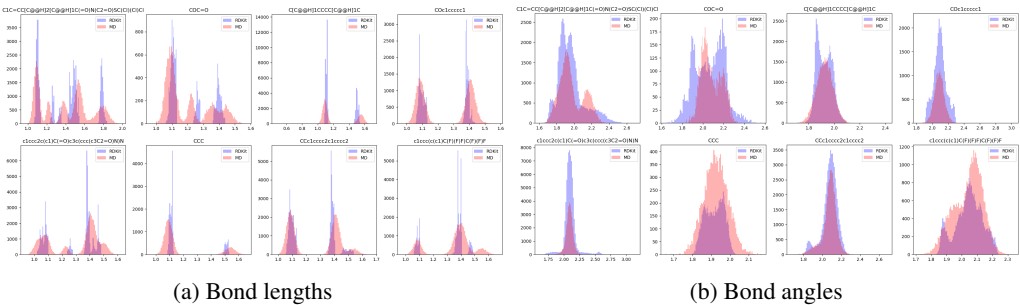

(a) Bond lengths           (b) Bond angles

Figure 5: Histograms of local structures of MD data (pink) compared to local structures generated by RDKit (purple) for the 8 molecules in our dataset. Left is bond lengths, right is bond angles. Note that RDKit significantly underestimated the vibrations of the bond lengths, resulting in peakier modes for all molecules.

### A.3.5   CORRELATION BETWEEN SAMPLING PROBABILITY AND REWARD

To estimate the sampling log-probabilities of the trained Torsional-GFN model, we follow the algorithm described in (Volokhova et al., 2024). We use $N = 10$ trajectories for each estimation.

$$\rho_{\log p_\top^\theta, logR} = \frac{\text{cov}(\log p_\top^\theta(\Phi_i|G, L), \log R(\Phi_i|G, L))}{\sigma_{\log p_\top^\theta(\Phi_i|G,L)} \sigma_{\log R(\Phi_i|G,L)}}. \tag{16}$$

To measure correspondence between sampling probabilities and rewards we use correlation coefficient of their logarithms as defined in Eq. 16. The choice of the logarithms instead of the direct values is motivated by the observation that logarithmic scale allows to mitigate the over-focus on the high-reward samples, while correlation of the direct values is often biased by the high-reward samples (see for example Figure 6), especially for the molecules with sharp changes in the energy landscape.

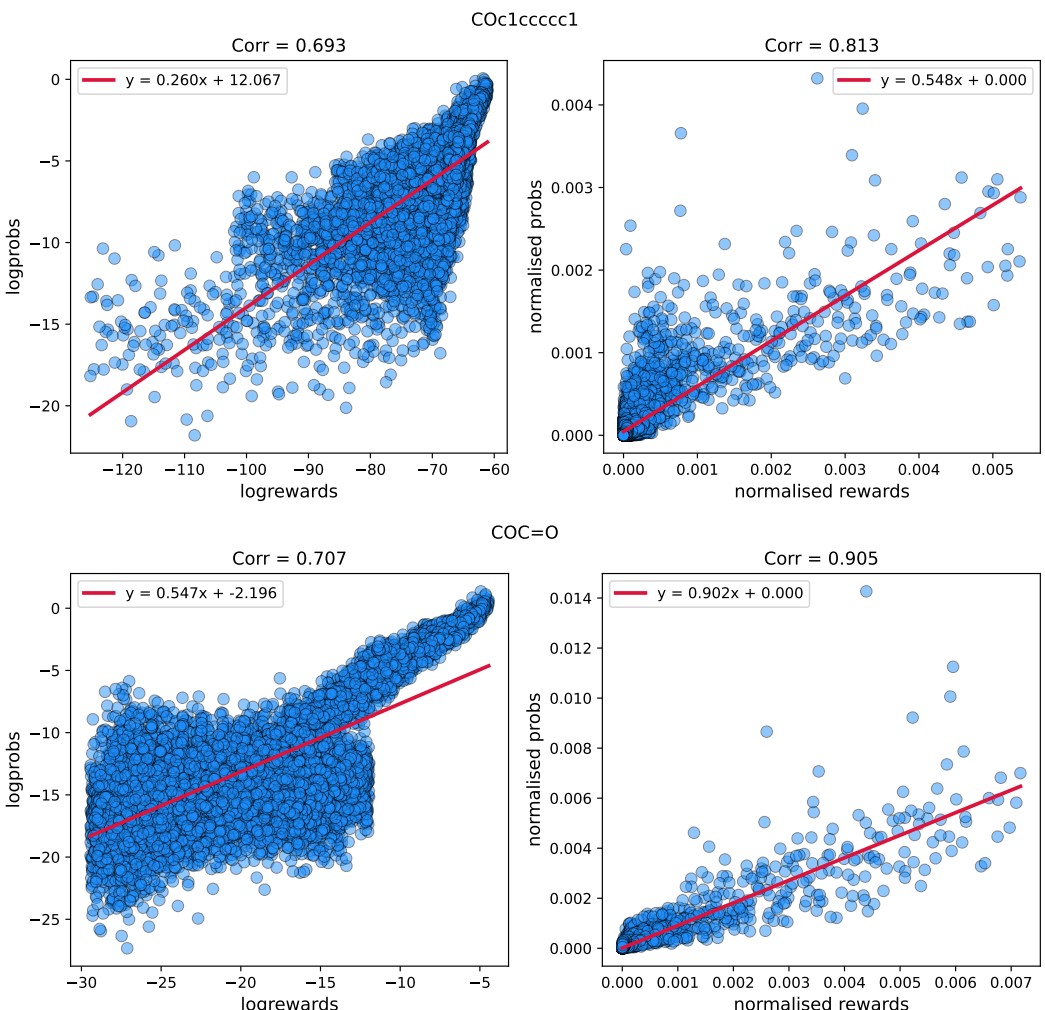

Figure 6: Examples of the scattered plots of rewards and sampling probabilities in the log scale and without it. The values are computed on a grid of 10,000 torsion angles values. As one can see, the correlation coefficient of the direct values is higher than for the logarithmic ones due to the low-reward values collapsing in the corner close to zero. This pattern shows that correlations of the direct values can underestimate the discrepancy of the sampling probability and reward in the low reward regions.

### A.3.6    2D VISUALISATIONS FOR TRAIN MOLECULES

This section shows additional visualisations of the 2D log-rewards alongside the learned distributions in Figures 7 to 11.

### A.3.7    2D VISUALIZATIONS FOR TEST MOLECULES

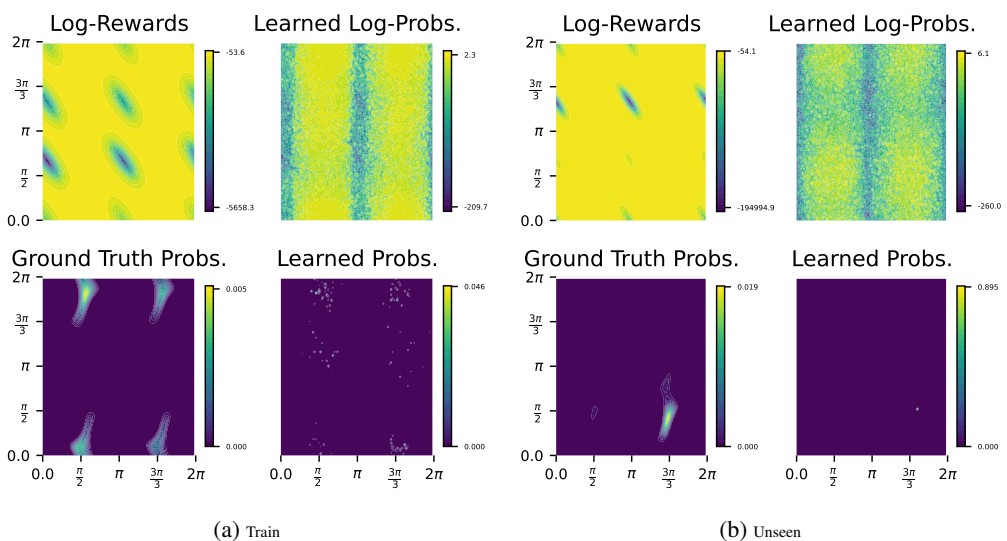

(a) Train

(b) Unseen

Figure 7: Visualization of the log-rewards and ground truth probabilities alongside the learned sampling distributions for molecule C1C=CC[C@@H]2[C@@H]1C(=O)N(C2=O)SC(Cl)(Cl)Cl in the train dataset, for train and unseen local structures.

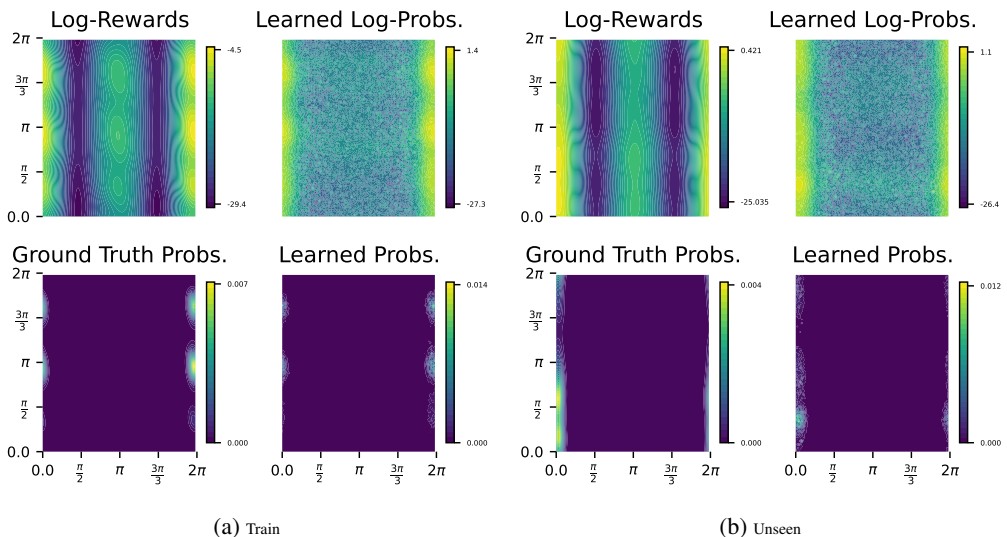

(a) Train

(b) Unseen

Figure 8: Visualization of the log-rewards and ground truth probabilities alongside the learned sampling distributions for molecule COC=O in the train dataset, for train and unseen local structures.

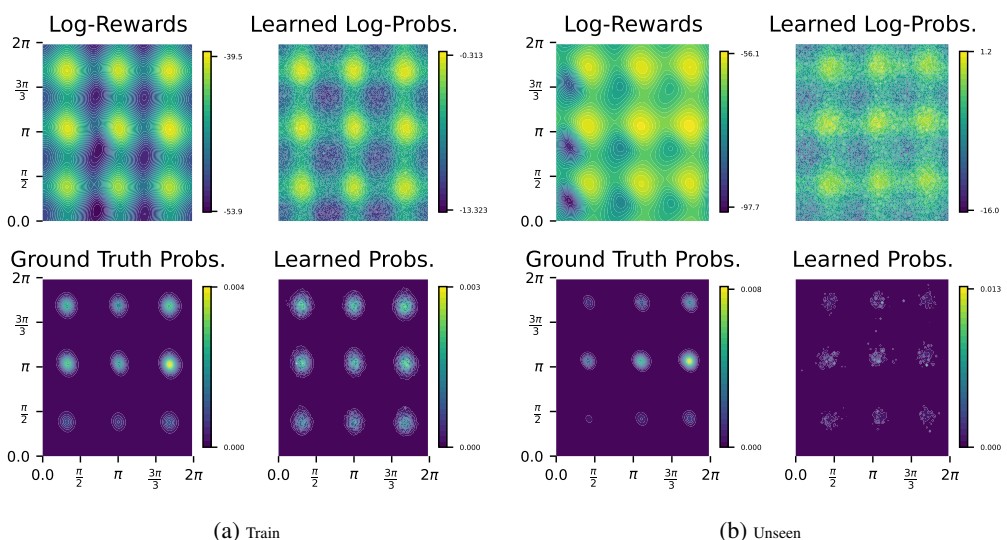

(a) Train

(b) Unseen

Figure 9: Visualization of the log-rewards and ground truth probabilities alongside the learned sampling distributions for molecule C[C@@H]1CCCC[C@@H]1C in the train dataset, for train and unseen local structures.

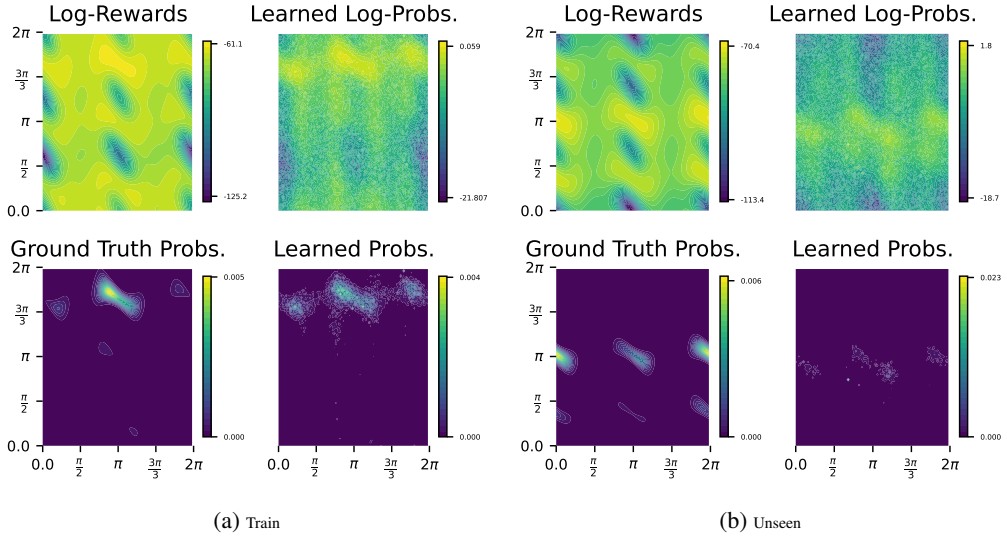

(a) Train

(b) Unseen

Figure 10: Visualization of the log-rewards and ground truth probabilities alongside the learned sampling distributions for molecule c1ccc2c(c1)C(=O)c3c(ccc(c3C2=O)N)N in the train dataset, for train and unseen local structures.

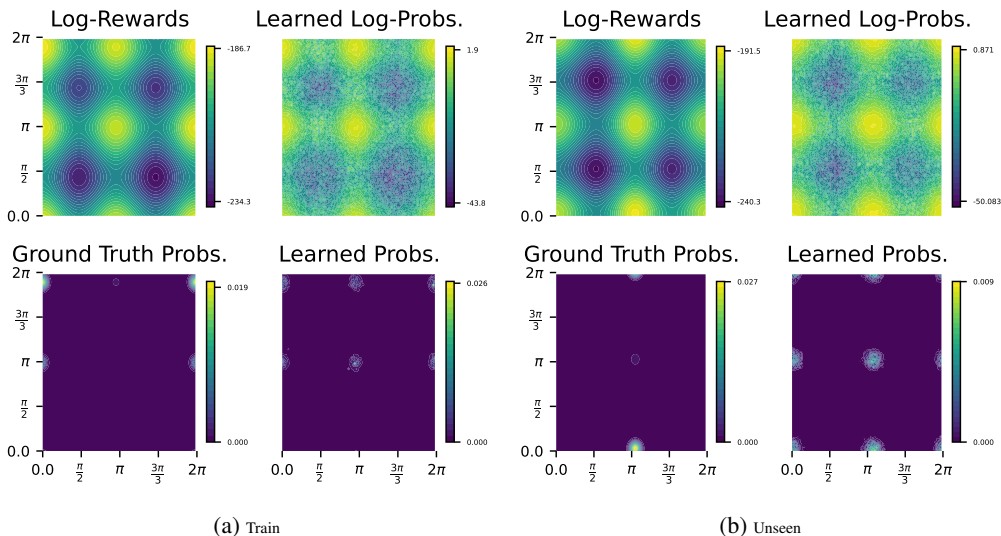

Figure 11: Visualization of the log-rewards and ground truth probabilities alongside the learned sampling distributions for molecule CCC in the train dataset, for train and unseen local structures.

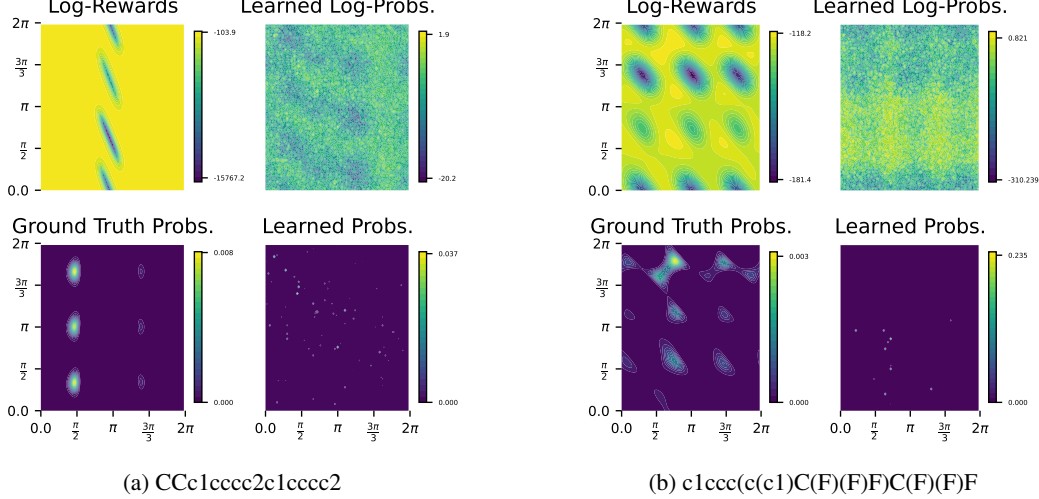

Figure 12: Visualization of the log-rewards and ground truth probabilities alongside the learned sampling distributions for 2 molecules in the test dataset.

