# OpenReview forum: "Torsional-GFN: a conditional conformation generator for small molecules"
_ICLR.cc/2026/Conference — Submitted to ICLR 2026_

### Official Review · Reviewer_Bmi7 · 2025-10-31

**Soundness:** 4
**Presentation:** 3
**Contribution:** 2
**Rating:** 2
**Confidence:** 4

**Summary:**

The authors introduce Torsional-GFN, a GFlowNet-based method to sample molecular conformations (and energies) proportional to the Boltzmann distribution. Unlike previous work, the authors introduce a equivariant graph neural network VectorGNN to allow for zero-shot generalization to new molecular graphs. Results show that Torsional-GFN samples conformations that somewhat match the true distribution obtained from MD, although for unseen graphs it is somewhat successful for one molecule and unsuccessful for the other.

**Strengths:**

1. The generation of molecular conformations maps very nicely to the structure of GFlowNets, so I think the approach used by the authors is quite reasonable given the problem setting
2. The presentation in the Methods is very clear, and the design choices are reasonable
3. The experiments show promising results on the train molecules, but performance on the test molecules is inconsistent

**Weaknesses:**

Compared to previous work (Lahlou et al 2023, Volokhova et al 2024), it seems that the main novelty of Torsional-GFN is the ability to generalize to unseen molecular graphs at inference time. However, the results on unseen test molecules are very inconsistent, with one molecule performing okay and the other having no correlation with the true distribution. Therefore, it is difficult to claim that the generalization to unseen test molecules works well, meaning in my opinion the paper does not achieve its main motivation. Beyond this main weakness, additional concerns include:

1. The authors claim that "if we care about the full potential energy landscape, [RDKit] approximations do not hold." However, I don't see how Appendix A.3.4 addresses this problem. A.3.4 includes data on the bond lengths not matching exactly with MD, but to really show that the approximation is inaccurate I'd like to see the energy landscape from RDKit plotted against that from MD.
2. There are no baselines besides MD. Including previous works that do Boltzmann sampling, such as Lahlou et al. 2023, would be informative as a point of comparison (although I understand that their model requires training for each new molecule). Other baselines that are applicable, I think, include Noe et al. 2019 and some of the diffusion models mentioned in the Related Work.
3. MD sampling is done in vacuum, which I believe is not really interesting for many drug discovery applications. I think simulations should be done in solvent.

**Questions:**

1. Did the authors confirm that the 1ps samples taken from the MD simulations are in fact uncorrelated? It's probably fine, but it would be nice to know that the samples are taken far enough apart to be uncorrelated.
2. How long does it take to run Torsional-GFN and get a good distribution of samples? How does this compare to the runtime of MD?
3. Why not compare GFlowNets with other generative models?

---

### Official Review · Reviewer_tbTh · 2025-11-01

**Soundness:** 2
**Presentation:** 3
**Contribution:** 2
**Rating:** 4
**Confidence:** 3

**Summary:**

This paper presents Torsional-GFN, a novel framework based on the GFlowNet paradigm designed to model molecular conformational spaces. Building on prior GFlowNet approaches, the authors introduce several methodological enhancements to improve scalability—most notably, replacing per-molecule GFlowNets with a single GFlowNet leveraging a new VectorGNN architecture. The model is trained on the FreeSolv dataset, and its performance is assessed through detailed analysis on a validation set.

**Strengths:**

While the use of GFlowNets for conformation generation is not entirely new, the scaling of GFlowNets to a unified model across molecules represents a moderately significant contribution.

The authors improve both the theoretical formulation and architectural design of the framework, proposing meaningful modifications to make GFlowNets more suitable for this problem.

The paper is clearly written and reasonably easy to follow, with good explanations of the methodology and analysis of generated conformations relative to the ground truth distribution.

**Weaknesses:**

Despite promising direction, the practical evaluation remains limited and raises questions regarding the framework's scalability and general applicability.

Although scalability is claimed as a key advantage (“our work presents a promising avenue for scaling the proposed approach for larger molecular system”), experiments are conducted on only 8 molecules (6 from training and 2 from testing). This small sample size does not convincingly support claims about scalability—in terms of either dataset size or molecular complexity. It is unclear from the paper what factors currently limit training and evaluation on larger datasets or larger molecules.

The experimental section focuses on internal analysis of the proposed model but omits comparisons with prior GFlowNet-based conformation models (e.g., [a]) and alternative torsion-angle–based conformation generation methods (e.g., [b], [c]). Without such comparisons, it is difficult to properly assess performance improvements or the practical advantages of the proposed approach.

The paper would be significantly strengthened by either including baseline comparisons on a larger benchmark dataset (e.g., GEOM-DRUGS [d]), or adding a dedicated section detailing the current limitations that prevent scaling to such benchmarks, including computational, architectural, or training challenges.

a. Towards equilibrium molecular conformation generation with GFlowNets, Volokhova et al.

b. Torsional Diffusion for Molecular Conformer Generation, Jing et el.

c. COSMIC: Molecular Conformation Space Modeling in Internal Coordinates with an Adversarial Framework, Kuznetsov et al.

d. GEOM, energy-annotated molecular conformations for property prediction and molecular generation, Axelrod et al.

**Questions:**

My main concerns regarding model applicability and scalability are already outlined in the Weaknesses section. Clarification on the following points would be particularly helpful:

What are the primary computational or methodological barriers preventing application to larger datasets or larger molecular systems?

Are there plans or proposed modifications that could prove or enable scaling to datasets such as GEOM-DRUGS?

---

### Official Review · Reviewer_w3qE · 2025-11-01

**Soundness:** 3
**Presentation:** 3
**Contribution:** 2
**Rating:** 4
**Confidence:** 4

**Summary:**

This work introduces Torsional-GFN, a conditional GFlowNet sampler that generates molecular torsion angles based on the molecular graph and local geometry from MD (bond lengths and angles). It approximates the Boltzmann distribution over conformations by using energy as the reward. A GNN policy outputs the parameters of a von Mises mixture for torsion updates. The model is trained on six molecules and tested on two others, each with two rotatable bonds, demonstrating transfer to unseen local environments and limited generalization to new molecules.

**Strengths:**

- Introduces forward and backward policies based on a von Mises mixture, with well-defined training objectives.
- Shows that the conditional sampler adapts to unseen local environments, indicating potential for amortized conformer generation workflows.

**Weaknesses:**

- While the combination of "torsional neural sampler with GFlowNet for molecules" is new, the work itself is primarily a combination of existing tasks and methods: molecular torsional neural sampler is already formalized in Adjoint Sampling (ICML 2025) and ASBS (NeurIPS 2025); molecular sampling with GFlowNet with similar goals have been explored in the cited works and more (e.g., arXiv:2505.19552), while with different training objectives.
- Compared to amortized conformer generation in AS/ASBS on the GEOM dataset (few thousand molecules), the benchmark with 6/2 train/test molecules with two torsions limits the strength of claims about generalization and scalability.
	- I note that large-scale evaluation in previous works are supported by industry-level compute, so matching the "size" of the benchmark might not be feasible. But adding a comparison against diffusion-based samplers in the same setting (for example, training AS on this task) would provide insights on the differences between sampling algorithms.
- Training requires >40 GB GPU even for small molecules with classical force field rewards, which may hinder practical adoption.

**Questions:**

- Reference samples come from MD with OpenFF, while rewards use MMFF94s. This mismatch can confound comparisons between reference and generated energy distributions. Please justify the choice or use a coherent force field for both MD and reward.
- JSD^p and rho values are compared only against energy-derived targets. Can you also report these metrics against MD torsion distributions?
- Please compare VectorGNN performance and speed to other equivariant GNN baselines when used as the policy backbone.
- It would be nice to bring Fig. 1 to the main text to show an overview of the method.

---

### Official Review · Reviewer_bVCV · 2025-11-01

**Soundness:** 3
**Presentation:** 3
**Contribution:** 3
**Rating:** 6
**Confidence:** 2

**Summary:**

This paper introduces Torsional-GFN, a novel method for generating molecular conformations from the Boltzmann distribution. The approach leverages a conditional GFlowNet to sample torsion angles and demonstrates its zero-shot generalization capability, particularly to unseen local structures (bond lengths and angles). This is a highly relevant and timely research direction, applying the emerging GFlowNet framework to the important problem of molecular conformation generation.

**Strengths:**

1.  The paper begins by clearly articulating the importance of efficiently sampling molecular conformations from the Boltzmann distribution and its applications in drug discovery. It accurately identifies the limitations of existing methods (e.g., computational cost of MD and data dependency of diffusion models) and makes a compelling case for GFlowNets as an alternative that can be trained solely on a reward function (energy).
2. The paper includes multiple metrics and good visualizations.

**Weaknesses:**

1. The dataset is in very limited scale. There is no demonstration of scalability to larger molecules or varying numbers of torsion angles.
2. One test molecule shows very poor performance and the other test molecule shows "limited coverage of some modes". This suggests generalization to unseen molecules is not reliably achieved.
3. No comparison with recent diffusion-based methods (TorsionalDiffusion, etc.) or other generative approaches.
4. In Equation (2), the authors simplify the problem by setting the volume change factor √det(g) to 1. The authors should discuss the practical impact of this approximation more deeply. Simply stating "future work will consider it" is insufficient.
5. The VectorGNN is first pre-trained on a supervised task. This is good engineering practice, but it obscures the source of the final model's performance.

**Questions:**

1. How does the method scale to molecules with 5, 10, or 20+ torsion angles?
2. Can the authors provide comparison with recent diffusion-based methods (TorsionalDiffusion, etc.) or other generative approaches.
3. The paper sets √det(g) to 1 for simplicity. Could the authors quantify or discuss the potential bias introduced by this approximation?
4. Could the authors provide an ablation study to isolate the contribution of the blocks of the framework?

---

### Meta-Review · Area_Chair_jXuB · 2026-01-07

**Summary:**

The paper proposes an approach for generating molecular conformations from the Boltzmann distribution. The approach is based on the intrinsic coordinate representation, i.e., bond lengths and angles and torsional angles, and the model is based on GFlowNet for the tortional angles and is trained with energy function as a reward.

Reviewers acknowledge the exploration of another technical approach using GFlowNet for molecular structure sampling. Nevertheless, they also expressed concerns about the significance and quality of the work:
1. The experiments are limited in scale, in terms of both the size of considered molecules and the number of molecule species, especially considering size of common public MD datasets. The experimental results are neither consistently supportive.
2. No relevant baseline that supports the claim of conceptual advantage is presented.
3. The novelty of the work is not ground-breaking, and it still requires MD simulation for sampling bond lengths and angles before the model can run inference on test cases, and this type of structural representation would restrict it to organic molecules.

**Reviewer Concerns:**

The authors did not post a rebuttal. The three common concerns among the reviewers listed above are reasonable challenges. In all, the paper could be an approach to scale up generative model training for molecular conformation ensemble sampling only from the energy function by reducing the degrees of freedoms, but the quality of finish is not satisfying.

**Reviewer Scores:**

The authors did not post a rebuttal.

---

### Decision · Program_Chairs · 2026-01-26

Reject